# Performance of the 5th Generation Indoor Wireless Technologies-Empirical Study

**Mika Hoppari \*** 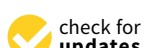**, Mikko Uitto, Jukka Mäkelä, Ilkka Harjula and Seppo Rantala**

VTT Technical Research Centre of Finland Ltd., 90540 Oulu, Finland; mikko.uitto@vtt.fi (M.U.); jukka.makela@vtt.fi (J.M.); ilkka.harjula@vtt.fi (I.H.); seppo.j.rantala@vtt.fi (S.R.)
\* Correspondence: mika.hoppari@vtt.fi

**Abstract:** The evolution of 5th generation (5G) cellular technology has introduced several enhancements and provides better performance compared to previous generations. To understand the real capabilities, the importance of the empirical studies is significant to also understand the possible limitations. This is very important especially from the service and use case point of view. Several test sites exist around the globe for introducing, testing, and evaluating new features, use cases, and performance in restricted and secure environments alongside the commercial operators. Test sites equipped with the standard technology are the perfect places for performing deep analysis of the latest wireless and cellular technologies in real operating environments. The testing sites provide valuable information with sophisticated quality of service (QoS) indicators when the 5G vertical use cases are evaluated using the actual devices in the carrier grade network. In addition, the Wi-Fi standards are constantly evolving toward higher bit rates and reduced latency, and their usage in 5G dedicated verticals can even improve performance, especially when lower coverage is sufficient. This work presents the detailed comparative measurements between Wi-Fi 6 and 5G New Radio (NR) performance in indoor facilities and extensive results carried out in 5G and beyond test site located in Finland. The results gathered from the extensive test sets indicate that the Wi-Fi 6 can outperform the 5G in the indoor environment in terms of throughput and latency when distance and coverage do not increase enormously. In addition, the usage of wireless technologies allows improved uplink performance, which is usually more limited in cellular networks. The gained results of our measurements provide valuable information for designing, developing, and implementing the requirements for the next-generation wireless applications.

**Keywords:** wireless; 5G NR; Wi-Fi 6; latency; throughput

## 1. Introduction

While research is already taking it first steps towards sixth generation (6G) networks, 5G connectivity has started to take place in actual commercial networks and research test beds around the globe, of which the 5G and beyond test network in Finland [1] is one of the infrastructures. The validation of the 5G use cases through information gathered from extensive field and laboratory measurements with actual hardware, software, and equipment can provide valuable knowledge for future development and implementation for 5G technology. The evaluation and measurements concluded in this paper using the latest Wi-Fi and 5G communication technologies already reveal the performance and limitations for their use in real operating environments. As this paper illustrates with live measurements conducted in our 5G test network, the performance of current wireless technologies is sufficient to meet both latency and data rate requirements.

5G evolution has enabled transmissions with a data rate with enhanced mobile broadband (eMBB), and ultra-reliable low-latency communications (URLLC) for decreasing the network delay. These elements are critical when the quality of experience/service (QoE/QoS) based services are designed in a wireless environment relying only the available cellular

connection with high reliability. The 5G network slicing feature [2,3] enables prioritized services for improving reliability. The services can contain critical and highly prioritized data, whose reception in the terminal should be flawless within the expected time frame. In many industrial wireless applications, delay jitter is more important than the average delay, because delay fluctuations can be more severe for the time-sensitive applications.

The huge leap in performance made possible by 5G will allow network operators to offer wireless broadband as a real alternative to people's primary internet connection, together with traditional broadband technologies. However, Wi-Fi development has evolved rapidly alongside the cellular standardization, and the usage of Wi-Fi in some use cases can be even more beneficial. The development of Wi-Fi 6 and Wi-Fi 6E sees Wi-Fi as the most efficient and noteworthy way to connect an ever-increasing number of home devices such as computers, televisions, phones, and smart speakers [4]. Similarly to in 5G network slicing, Wi-Fi 6 possesses slicing feature for enabling flexible radio resource management for specific applications. Aruba Air Slice [5] is one of the examples. The development of Wi-Fi 7 is already in the pipeline, aiming at 4.8 times higher data rates than Wi-Fi 6 [6]. Thus, it is foreseen that cellular technologies will dominate the use cases related to wide-area coverage, whereas Wi-Fi technologies are more suitable for indoor scenarios due to their lower deployment costs [7].

The usability of 5th-generation technology is being approached through various vertical sectors. The 5G Infrastructure Public Private Partnership (5G PPP) [8] defines the following main verticals: Automotive, Manufacturing, Media, Energy, eHealth, Public Safety, and Smart Cities. In order to understand the real capabilities and limitations of the technology, several pilots have been realised. For automotive, there have been studies related to connected and autonomous cars [9–11]. Manufacturing sector hosts huge opportunities and studies, such as [12–15]. Video broadcasting and delivery studies [16] have paved the way for the scalable high-quality media delivery solutions. The latest studies on the eHealth sector [17] have introduced new innovation solutions in order to increase the quality of life that can benefit from 5G technology. Smart energy grids and cities, together with public safety solutions [18–20], are showing very promising results towards a more reliable and secure society for the future. What is common in most of the vertical sectors is the clear benefit of higher capacity and better performance introduced by the latest wireless technologies.

As the 5G and beyond network in Finland holds the capability of testing and evaluating both 5G and Wi-Fi in terms of network performance, we carried out an extensive set of capacity measurements from several indoor operation points. The earlier comparative performance studies [21,22] as well as [23] are based more on simulations and modelling 5G rather than performing actual field measurements. The results presented later in this paper help us to identify the possible bottlenecks and pain points related to utilization of the wireless technologies in the various use cases presented by the industry sectors. One of our ongoing projects focuses specifically on indoor remote operations, where a deep understanding of wireless technology and technology limitations are significant.

This paper is organized in the following way. Section 2 discusses the architecture and theoretical performance of the latest Wi-Fi and 5G technologies in order to justify better their comparison with each other. Section 2 presents the evaluation environment, setup, and tools. Section 3 illustrates the results of our measurements, and finally, in Section 4, conclusion is drawn.

## 2. Materials and Methods

In this section, we highlight the theoretical background related to performance behind 5G and Wi-Fi technologies and discuss their equivalent comparison with each other. In addition, a detailed description of the evaluation setup with the related infrastructure is presented.

### 2.1. Theoretical Performance of 5G and Wi-Fi 6 Links

Calculating the throughput for both 5G New Radio (NR) and IEEE 802.11ax single-user links can be done by using the formulas presented later in this section. These calculations are based on sets of parameters that are chosen to represent the desired radio configuration, and the outcome reflects the throughput that could be achieved with these parameters. However, it has to be noted that although we get a theoretical maximum throughput for the given parameter configuration, in practice, there are many limitations to how the parameters can be set, from the channel quality to the boundaries set by the regulators.

The equation for calculating 5G NR throughput can be found in 3GPP TS 38.306 [24]. The approximate data rate $d$ (in Mbps) for a given number of aggregated carriers in a band is computed as

$$d = 10^{-6} \sum_{j=1}^{J} \left( (v_{\text{layers}}^{j} Q_{m}^{j} f^{j} R_{\text{max}}) (\frac{12 N_{\text{PRB}}^{\text{BW}(j)\mu}}{T_{S}^{\mu}})(1 - \text{OH}^{j}) \right), \quad (1)$$

where J is the number of aggregated component carriers in a band or band combination. $R_{\text{max}} = 948/1024$, and for the $j$th component carrier, $v_{\text{layers}}^{j}$ is the maximum number of supported layers; $Q_{m}^{j}$ is the maximum supported modulation order; $f^{j}$ is the scaling factor, which can take the values 1, 0.8, 0.75, and 0.4; $\mu$ is the numerology value defined in [25]; $T_{S}^{\mu}$ is the average orthogonal frequency-division multiplexing (OFDM) symbol duration in a subframe; $N_{\text{PRB}}^{\text{BW}(j)\mu}$ is the maximum resource block (RB) allocation in bandwidth BW($j$); and OH$^{j}$ is the overhead, which can take value 0.14, 0.18, 0.08, or 0.1, depending on the used frequency range. Full details on the calculation of the parameter values can be found in [24].

For IEEE 802.11ax, the maximum link throughput formula can be found, for example in [26]:

$$d = R_{c} \log_{2}(Q_{m})(N_{\text{data}}^{\text{BW}} t_{s} N_{\text{SS}}) \quad (2)$$

where $R_{c}$ is the coding rate (maximum being 5/6), $Q_{m}$ is the maximum supported modulation order, $N_{\text{data}}^{\text{BW}}$ is the number of data subcarriers as a function of bandwidth, $t_{s}$ is the time per OFDM symbol (13.6 μs with 800 ns guard interval), and $N_{\text{SS}}$ is the number of spatial streams ranging from 1 to 8.

The maximum available modulation order and coding rate depend on the received signal strength, and vast majority of the other parameters have been set to certain values at the implementation phase. For the 5G, the parameters affecting the practical performance of a link are the available bandwidth, the number of supported spatial layers, and the slot format that defines the uplink/downlink (UL/DL) ratio of the data transmission. By assuming 40 MHz bandwidth at 3540 MHz carrier frequency, 256 quadrature amplitude modulation (QAM), 4 spatial layers, and 1/4 UL/DL ratio, we achieve a maximum throughput of 698 Mbps for DL and 186 Mbps for UL (total sum 884 Mbps), respectively.

Similarly, the defining parameters for the Wi-Fi 6 link are the available bandwidth and the number of spatial streams. By assuming 40 MHz bandwidth in 5 GHz center frequency and assuming also four spatial layers and 256 QAM, we get a maximum throughput of 917 Mbps.

From the above simple calculations, we can see that with alike parameters, both systems produce a comparable throughput. The practical throughput in the field depends on the two main factors: how much resources are available for the system and how well the system can utilize them. There are two major differences between the systems. First, 5G NR operates on dedicated bands that are reserved solely on the specific purpose, while Wi-Fi 6 operates mainly on the Industrial, Scientific and Medical (ISM) radio bands, where there may also be other operating devices. Secondly, 5G NR uses scheduled access where the medium access control (MAC) algorithm determines the usage of the RBs, while Wi-Fi 6 is based on contention where data can be transmitted only after certain contention criteria are met.

For 5G, regulators have assigned frequency in three broad ranges: high bands (e.g., mmWave), mid bands (e.g., 1–10 GHz), and low bands (e.g., below 1 GHz). These serve different purposes. The high bands support the fastest throughput, the mid bands offer a good mixture of coverage and capacity, and the low bands help to provide strong wide area and in-building coverage [27]. So far, the focus has been on the 3.5 GHz range to support initial 5G launches, followed by the mmWave awards in the 26 GHz and 28 GHz bands. For example in Finland, the mmWave band in 26.1–27.5 GHz has been divided between the three operators [28].

For Wi-Fi 6, the frequency landscape looks different from that for the 5G. Wi-Fi 6 is a term that covers IEEE 802.11ax-2021 standard operating around 2.4 and 5 GHz carrier frequencies, and Wi-Fi 6E refers to the same standard operating at 6 GHz carriers [29]. The ISM band at 24 GHz remains vacant from Wi-Fi systems, but it has been considered as a candidate band for Broadband Wireless Access (BWA) systems in the USA and Canada [30]. The competitor for the 5G operating in the mmWave region is Wi-Fi standard IEEE 802.11ad from 2009, which operates at 60 GHz carrier frequency, offering data rates up to 7 Gbit/s.

*2.2. Evaluation*

In this subsection, we highlight the test infrastructure for the evaluation and provide the essential parameters regarding the network setup and configuration. In addition, the methods associated with the tools for performing the measurements are illustrated.

### 2.2.1. 5G and Beyond Test Network

Test were made at 5G and Beyond Test Network, operated by VTT Technical Research Centre of Finland, which offers tens of different types of standard radio access nodes including latest 5G NR as well as Wi-Fi and cellular IoT technologies complemented with non-terrestrial satellite connections. The test network consists of several private 5G sites with full-scale end-to-end (E2E) data communication system supporting 5G standalone (SA) and non-standalone (NSA) architectures. For dedicated measurements, it is possible to isolate part of the test network resources and equipment to be utilized in dedicated tests and development projects, similarly to how it was done also with the study presented in this paper. Figure 1 shows the architecture that was used in this study. The figure illustrates the network setup, which consists of non-standalone 4G (evolved node B, eNB) and 5G NR (next-generation node B, gNB) base stations as well Wi-Fi 6 access points providing the radio coverage. Wi-Fi network coverage was not integrated to the 3GPP 5G architecture; instead, it was a dedicated test setup utilizing the same backbone network as the 5G core and multi-access edge computing (MEC) servers. In the 5G core network, backbone 5G core network instances together with MEC support were used. The MEC application server was realized by following the ETSI specified distributed EPC architecture specification [31].

### 2.2.2. Evaluation Setup and Tools

The detailed measurements and extensive results were carried out in 5GTN in comparison with Wi-Fi 6 and 5G NR performance in indoor facilities. The evaluation was made by making throughput and delay measurements with different wireless connection techniques. The 5G measurements were made by using Nokia 5G small cell base station with 60 MHz channel bandwidth. Wi-Fi 6 measurements were carried out with the Aruba 510 Series Wi-Fi 6 indoor access point (AP-515) [5] with 160 MHz channel bandwidth.

The measurements were made by anchoring each base station in turn to the other end of the office building and performing throughput and delay measurements at the specified measurement points. The Figure 2 shows the location of the base station and measurement points in the floor plan of the office building.

The measurement device was a Samsung S10 5G (2019) Android phone that had Keysight Nemo Handy [32] installed. Nemo Handy is an application that enables measuring wireless diagnostics information from the air interface. It provides several integrated tools for measuring the performance of telecommunication networks. In our test campaign,

we used the iPerf3 software to measure throughput and the Ping tool to measure bidirectional delay. The measurement results recorded by Nemo Handy were analyzed on a laptop with Nemo Outdoor software [33].

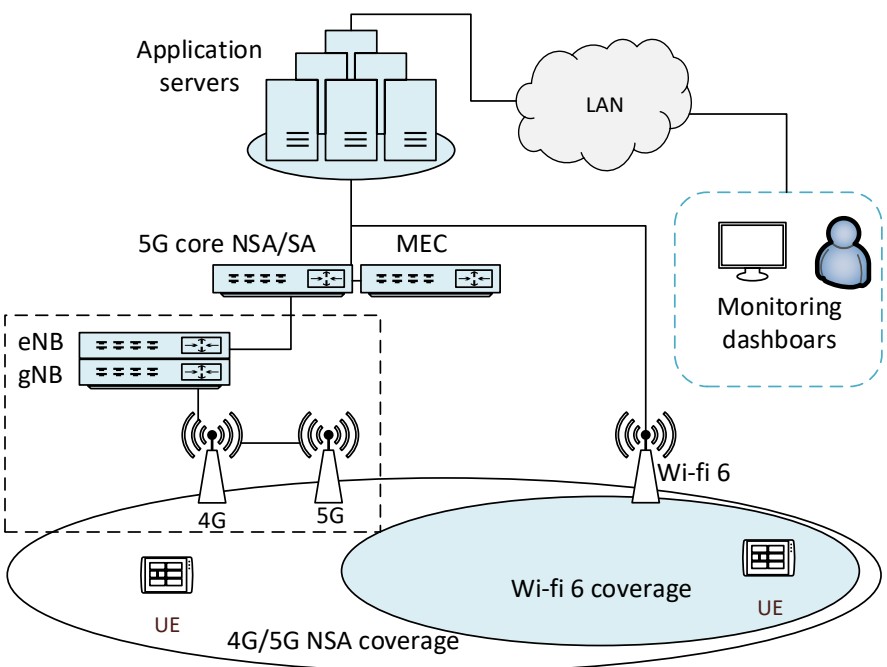

**Figure 1.** Architecture of the evaluation setup.

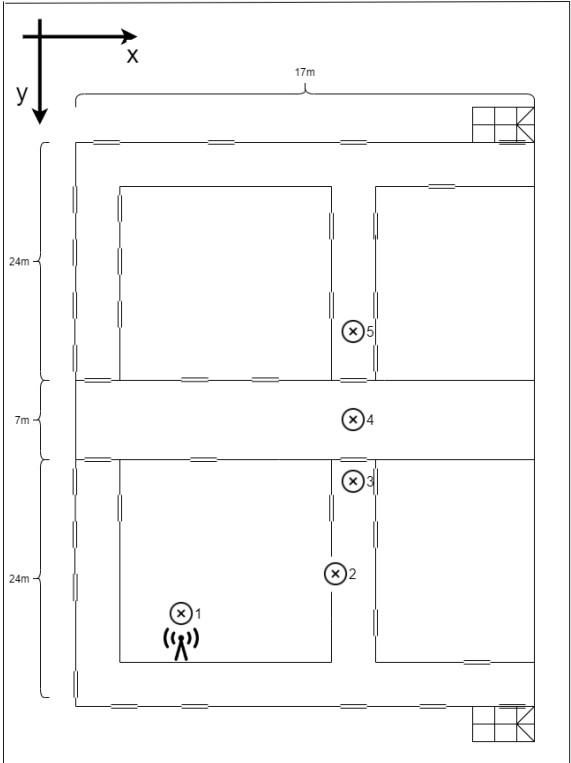

**Figure 2.** The measurement points presented on a map of VTT's office building.

Packet Inter Net Groper (PING) is a very simple transmission control protocol/Internet protocol (TCP/IP) utility [34]. Here, it is used to measure round trip time (RTT) between a

client and a server connected via 5G or Wi-Fi 6. Ping determines the RTT by sending one or more IP packets to the destination host and measuring the time it takes from sending the data packet to receiving an acknowledgment for that packet. iPerf3, on the other hand, is a tool for measuring the peak throughput of IP networks [35]. iPerf3 provides extensive support for setting parameters related to protocols, timings, and buffers.

Table 1 lists all the parameters that were used in the Ping and iPerf3 measurements. To eliminate instantaneous minimum and maximum values, Ping and iPerf3 measurements are repeated numerous times at each measurement point, and the results were averaged. During the Ping run, the RTT was measured with a 32-byte packet 600 times every 100 ms, and iPerf3 measured the throughput per second for 5 min. As noted in [36], the lack of packet order control of the user datagram protocol (UDP) results in the fact where throughput of TCP traffic is less than the UDP throughput. Thus, iPerf3 measurements of this evaluation were performed with the UDP protocol, which easily filled the 5G and Wi-Fi 6 links due to the lack of flow control. The evaluation of other protocols, such as the general-purpose (QUIC) communication protocol was left outside the scope of this paper.

**Table 1.** Ping and iPerf3 parameters.

| Ping | | iPerf3 | |
|---|---|---|---|
| Repeats | 600 | Duration | 300 s |
| Packet Size | 32 B | UDP Datagram Size | 1440 B |
| Interval | 100 ms | UDP Target Bitrate | 1 Gbps |
| | | Buffer Length | 410 KB |

### 2.2.3. Network Configuration

The measurement campaign included throughput and delay measurements for the Aruba Wi-Fi 6 access point and the Nokia 5G small cell indoor base station. Table 2 lists the most important parameter values for both the base stations. The 5G base stations used 3540 MHz carrier frequency, 60 MHz bandwidth, QPSK/4PSK modulation, 4 × 4 closed-loop spatial multiplexers, and 21 dBm maximum transmit power. The Wi-Fi 6 base station used 5150–5850 GHz carrier frequency, 160 MHz bandwidth, 4 × 4 MIMO, High Efficiency (HE) 20/40/80/160 modulation, and 24 dBm transmission power. The carrier aggregation (CA) technique, where multiple frequency blocks are assigned to the same user, was disabled in the 5G access point. By enabling the CA, the maximum possible data rate per user can be increased.

**Table 2.** 5G and WLAN configurations.

| Parameter | 5G | Wi-Fi 6 |
|---|---|---|
| Carrier frequency | 3540 MHz | 5.150 to 5.250 GHz 5.250 to 5.350 GHz, 5.470 to 5.725 GHz, 5.725 to 5.850 GHz |
| midrule Bandwidth | 60 MHz | 160 MHz |
| Maximum transmit power | 21 dBm | 24 dBm |
| MIMO mode | 4 × 4 Closed Loop Spatial Multiplexing | 4 × 4 MIMO |
| Modulation Scheme | Automatic | HE 20/40/80/160 |
| Carrier Aggregation | Disabled | - |
| ullaDeltaSinrMax | 15 | - |

## 3. Results

This section presents and compares the measurement results obtained with 5G and Wi-Fi 6 base stations. Table 3 presents all the measured values in all the measurement points illustrated in Figure 2. Throughput values are given as peak values and delay as an average value. Figures 3 and 4 compare the measured downlink and uplink throughputs between 5G and Wi-Fi, and Figure 5 compares delay. For 5G, values for both 1/4 and 3/7 DL/UL ratios are shown, which correspond the Frame Structure Type (FST) for allocating bandwidth to DL and UL.

We can also compare the measured throughput in Table 3 to the theoretical throughput by inserting parameter values from Table 2 into Equations (1) and (2). With these parameters and using FST 1/4 for 5G, the theoretical throughput for 5G DL and UL are 2218 and 222 Mbps and for Wi-Fi 9603 Mbps, respectively. We can see that for either system, the practical performance does not reach the theoretical throughput. Our interpretation is that the peak modulation and coding rates probably cannot be applied due to path loss and fading, and the spatial dimension remains under-utilized compared to the theoretical maximum. However, the results follow the theory in the sense where Wi-Fi, UL, and DL throughput are comparable in performance, while 5G follows the asymmetrical DL/UL resource allocation.

**Table 3.** Measured peak throughputs (Mbps) and average delay (ms) for Wi-Fi 6 and 5G (DLT/ULT = downlink/uplink throughput).

| Measurement Point | 5G | | | | | | Wi-Fi 6 | | |
| | fst 3/7 | | | fst 1/4 | | | | | |
| | DLT | ULT | Delay | DLT | ULT | Delay | DLT | ULT | Delay |
|---|---|---|---|---|---|---|---|---|---|
| 1 | 470 | 70 | 11 | 579 | 31 | 13 | 698 | 677 | 3 |
| 2 | 433 | 64 | 11 | 611 | 40 | 13 | 491 | 514 | 3 |
| 3 | 331 | 48 | 11 | 435 | 15 | 13 | 119 | 117 | 3 |
| 4 | 256 | 46 | 11 | 322 | 18 | 13 | 41 | 41 | 21 |
| 5 | 187 | 48 | 11 | 160 | 17 | 13 | 0 | 0 | 21 |

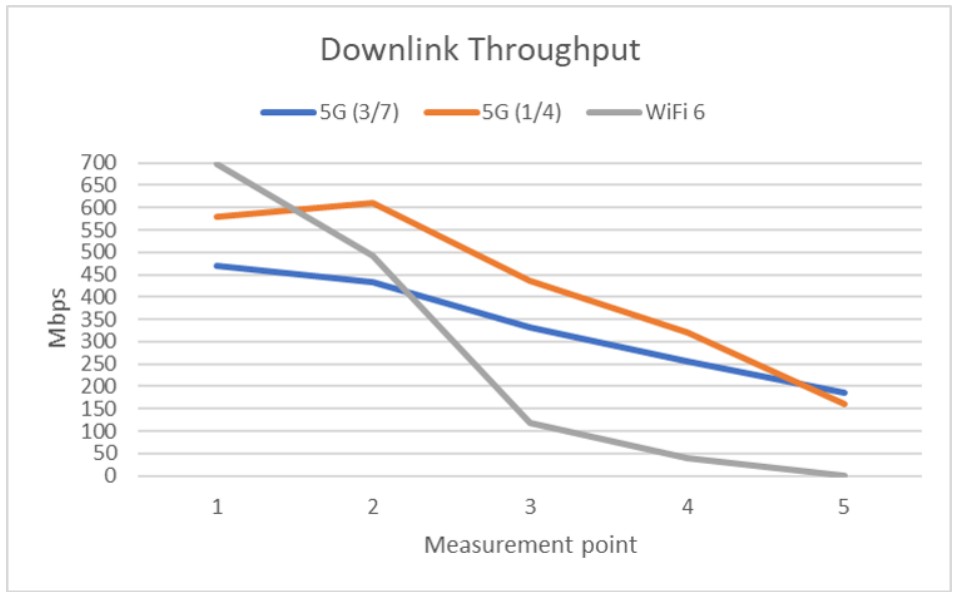

**Figure 3.** DL Peak Throughput.

Looking at Figures 3 and 4, it can be seen that Wi-Fi 6 outperforms 5G throughput both in uplink and downlink when the distance from the user equipment to the base station

is really short, but the situation changes rapidly as the distance to the base station increases. Similarly, the throughput measured with the 5G remains higher than with Wi-Fi 6 if there are signal blocking barriers between the user equipment and the base station, such as closed doors or walls. This is easily seen when comparing measurement points 3 and 4, which are located very close to each other but with a closed door between them. The results show that 5G achieved the best downlink throughput using frame structure type 1/4, but the best uplink throughput was measured using frame structure type 3/7.

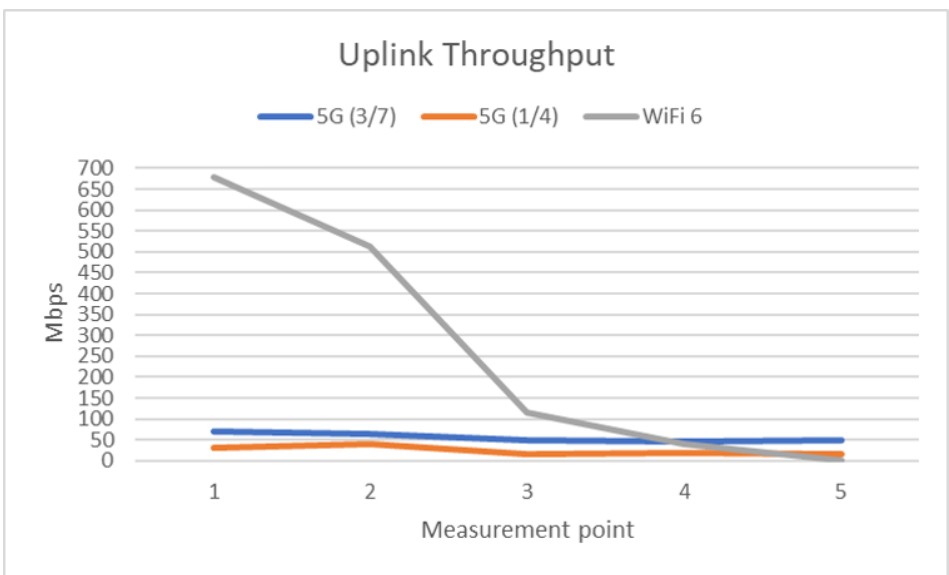

**Figure 4.** UL Peak Throughput.

The delay comparison shown in Figure 5 follows a similar trend as can be seen in the case of throughput measurements. Wi-Fi 6 is clearly ahead of 5G at close range, but as the distance increases, the delay of Wi-Fi 6 collapses while at 5G it remains stable. The collapse of the Wi-Fi 6 delay when moving from the measuring point 3 to the measuring point 4 is due to the fact that there is a closed door between those points, which clearly effectively blocks the communication between the Wi-Fi 6 base station and the user equipment.

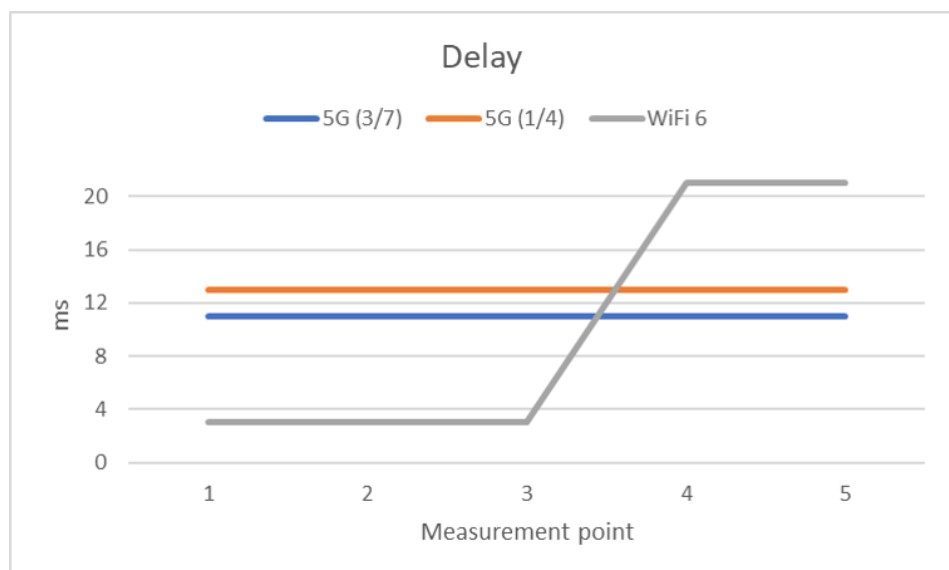

**Figure 5.** Average Delay.

## 4. Discussion and Future Work

The results gathered from the extensive test sets indicate that the state-of-the-art Wi-Fi 6 technology offers good enough performance and can even outperform the 5G NR in an indoor environment in terms of throughput and latency when distance and coverage do not increase enormously. Fifth-generation and beyond cellular networking seems to be more suitable for wide-area communications with extensive coverage and reliability alongside sufficient data rates and low latency. The usage of current Wi-Fi technologies allows especially improved uplink performance with low latency, which is critical for certain use cases and applications. Such use cases range, for example, from industrial automation to eHealth, which usually have either dedicated constant network environment. With upcoming 5G slicing technology, 5G radio access can be shared with public and private networks; in any case, it needs to be noted that Wi-Fi is also a good alternative technology for private network implementations, especially for the small scale networks. Because industrial factory sites can have multiple network connectivity needs ranging from visitors to automated robots, the slicing feature is needed to enable appropriate prioritized QoS management regardless of whether 5G or Wi-Fi is used.

The gained results from our experiments provide valuable information for designing, developing, and implementing the requirements for the next-generation wireless applications. Average delay under 5 ms enable basically real-time networking for remote operations and control and very low latency acts as an enabler for the safety-critical industrial applications [37]. Thus, the application delay can become the bottleneck for such use cases and should be therefore carefully designed and implemented according to the use case. Based on the results, there is no clear winner for the used technology, and both technologies have a road-map for future versions and are likely to coexist for a long time. The real choice between the Wi-Fi and 5G technology depends a lot on the use case, business models, and available frequency. The tests were run in a test network that has limited bandwidth available in the cellular frequencies, and the throughput performance between 5G and Wi-Fi 6 would be more comparable if the frequency bands available for the cellular operators could have been fully utilized in the tests. Authors will continue the evaluations studies as the technologies evolve, and the next step is to evaluate the radio technologies with higher frequencies above 24 GHz.

**Author Contributions:** The authors of this article worked as a team, and all had a significant contribution to the work performed, including the following: Conceptualization, J.M. and M.U.; methodology, I.H.; building the test setup and performing the actual test campaigns, M.H. and S.R. All authors have read and agreed to the published version of the manuscript.

**Funding:** This research has been funded and supported by VTT-Technical Research Centre of Finland Ltd. 5G and beyond test network has been supported also by Business Finland and Academy of Finland.

**Data Availability Statement:** Not applicable.

**Conflicts of Interest:** The authors declare no conflict of interest.

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
