# Peer review of "Performance of the 5th Generation Indoor Wireless Technologies-Empirical Study"

_futureinternet, doi:10.3390/fi13070180_

Round 1

Reviewer 1 Report

The paper work presents the detailed measurements and extensive results carried out in 5G and beyond test site located in Finland in comparison of Wi-Fi 6 and 5G New Radio (NR) performance in indoor facilities.

Several elements need to be corrected in the paper:

  • Please improve the quality of the figures (too small and quite illegible).
  • For Figures 3, 4, 5. The charts can be further improved in terms of the presentation of the results. Complete the names on the axes.
  • You can expand the summary of the paper and refer to other frequencies used in 5G.
  • There are several linguistic and stylistic errors. Please edit the text of the paper

Author Response

Rev1 /1: Base stations were used in the 5G test environment, but RRH was separated in Figure 1. Thus, the C-RAN architecture and the interface connecting the CU+DU (BBU) to the RU (RRH) were used. What split/option was used between the NG-RAN components and what might it mean when the BBU is located at a greater distance in the edge cloud? Note: the description of eNB/gNB in this context is inaccurate as RRH is a component of the eNB/gNB. Please have a discussion and describe in detail the influence of C-RAN architecture on latency.  The abstraction level of Figure 1 was modified according to the comments.  The deployment of 5G and LTE is not C-RAN even RRH is a separated hardware unit.  Evalution of different options for network deployment with C-RAN and D-RAN is a good comment and excellent topic for the future evaluations.

Rev1 /2: As part of the research, indoor propagation simulation (or analytical calculation according to the model have been chosen) adapted to the MIMO 4x4 system must be carried out, which will show the level of attenuation between the transmitter and receiver. On this basis, it will be possible to determine the selection of the modulation-code scheme (MCS) by the transmitter. A tabular summary of the sensitivity of the receivers used in the experiment should be presented here with regard to individual interfaces. This is a good idea and would give good insight how the individual measurements compare against theoretical values, but unfortunately out of scope for this publication given the timeframe. We have focused our efforts here on empirical evaluation of the performance.  

Rev1 /3: The split of functions performed by MEC and AP Wi-Fi is not clearly presented. Please describe this split in detail in relation to the possibility of functional split between BBU and RRH on the 5G link, if the C-RAN architecture is used. Description of the used MEC architecture was added to the text with the new reference to the ETSI distributed EPC architecture as well description for Wi-Fi integration. C-RAN architecture was not used.

Rev1 /4: There should be a discussion about the potential use of the QUIC protocol, instead of TCP/UDP, as it will be crucial in the near future and may significantly reduce the latency level in difficult propagation conditions. The impact of the QUIC protocol should be presented in relation to the techniques specific to the 5G-NR and 802.11ax interfaces. In this work we focused on TCP/UDP protocols and other protocols, such as QUIC, were left outside the scope of the topic.  

Rev1 /5: The authors wrote in the paper that the latency of a few milliseconds is sufficient for the controlled operation to work properly. How does this relate to the planned delay in the RAN domain below 1 ms in industrial networks and mIoT, and what delay did the authors actually obtain, because in Figure 5 an incorrect unit was used (probably copied from the previous graphs), which does not allow to determine the order of the data obtained? The errorneus figure labeling has now been corrected. We added  discussion on the need for low latencies to Discussion and future work section, and added a reference to a measurement-based paper that studies the applicability of URLLC on safety-critical processes.   

Rev1 /6: In the state-of-the-art part please include more references that will show the results of similar comparative research. We added three more references (20-22) of comparative studies regarding 5G and Wi-Fi performance evaluation referenced in the Introduction.

Reviewer 2 Report

General comments:

The presented paper is devoted to the well-known issues related to the measurement of throughput and latency in wireless networks. It is known that some time ago, the competition between 5G small-range networks and Wi-Fi networks operating in the 5GHz band began. Given that 5G systems can operate at the shared frequency bands, the threat of crowding out the Wi-Fi network is real. At this point, it should be added that the performance of comparative tests of these two technologies is justified. The results of tests carried out by the authors of the paper show that the performance of the interfaces is very similar, the same is shown by the results of simple calculations. The publication presents interesting test results, but in order to increase the value of the paper, some extensions and explanations should be added.

Detailed comments:

The paper was written in a concise and legible manner, but in order to increase its value, several extensions and corrections are required. Some dedicated comments:

  1. Base stations were used in the 5G test environment, but RRH was separated in Figure 1. Thus, the C-RAN architecture and the interface connecting the CU+DU (BBU) to the RU (RRH) were used. What split/option was used between the NG-RAN components and what might it mean when the BBU is located at a greater distance in the edge cloud? Note: the description of eNB/gNB in this context is inaccurate as RRH is a component of the eNB/gNB. Please have a discussion and describe in detail the influence of C-RAN architecture on latency.
  2. As part of the research, indoor propagation simulation (or analytical calculation according to the model have been chosen) adapted to the MIMO 4x4 system must be carried out, which will show the level of attenuation between the transmitter and receiver. On this basis, it will be possible to determine the selection of the modulation-code scheme (MCS) by the transmitter. A tabular summary of the sensitivity of the receivers used in the experiment should be presented here with regard to individual interfaces.
  3. The split of functions performed by MEC and AP Wi-Fi is not clearly presented. Please describe this split in detail in relation to the possibility of functional split between BBU and RRH on the 5G link, if the C-RAN architecture is used.
  4. There should be a discussion about the potential use of the QUIC protocol, instead of TCP/UDP, as it will be crucial in the near future and may significantly reduce the latency level in difficult propagation conditions. The impact of the QUIC protocol should be presented in relation to the techniques specific to the 5G-NR and 802.11ax interfaces.
  5. The authors wrote in the paper that the latency of a few milliseconds is sufficient for the controlled operation to work properly. How does this relate to the planned delay in the RAN domain below 1 ms in industrial networks and mIoT, and what delay did the authors actually obtain, because in Figure 5 an incorrect unit was used (probably copied from the previous graphs), which does not allow to determine the order of the data obtained?
  6. In the state-of-the-art part please include more references that will show the results of similar comparative research.

Summary:

The results of the experiments presented in the paper show the great potential of the 5G-NR and 802.11ax interfaces, which is undeniable. The evolution of the 802.11 interface towards introducing 4G/5G-like techniques indicates that competition is inevitable. The presented description of the experiment, after the above-mentioned corrections and extensions have been introduced, will constitute a good contribution to the knowledge of the future indoor use of wireless systems operating on commercial and shared frequency bands.

Author Response

Rev2 /1: Many grammatical errors through the whole manuscript. We went through the paper carefully and corrected the grammatical errors found.

Rev2 /2: The contribution of this manuscript is missing in section 1. How does this manuscript contribute to the state-of-the-art? Please add one paragraph about this. The state-of-the-art should be extended with relevant papers about 5G testbeds and papers comparing 5G and WiFi-6

We added three more references (20-22) of the comparative studies regarding 5G and Wi-Fi performance evaluation referenced in the Introduction.

Rev2 /3: Network slicing concept is important in 5G, even one factory may need slices with different requirements. Could you please add a paragraph reflecting your thoughts and finding in relation to potential use of WiFi-6 slicing? This is an important issue in future to be considered when selecting between cellular and Wi-Fi technologies. We added couple of references for 5G slicing in the Introduction and investigated the Air Slice feature in the used Wi-Fi access points. Analysis and thoughts were also added in the Discussion section.

Rev2 /3: The unit in Fig. 5 is not correct. The unit was corrected according to delay as “ms”.

Reviewer 3 Report

Many grammatical errors through the whole manuscript and even in the abstract such as
Abstract line 5, line 8
"Several test sites exists around the globe for introducing, testing, and evaluating new features, use cases, and ..."
"The testing sites provides valuable information with sophisticated quality of service (QoS) indicators ..."

The contribution of this manuscript is missing in section 1. How does this manuscript contribute to the state-of-the-art? Please add one paragraph about this.

The state-of-the-art should be extended with relevant papers about 5G testbeds and papers comparing 5G and WiFi-6 such as 
A. Zreikat, "Performance Evaluation of 5G/WiFi-6 Coexistence," International Journal of Circuits, December 2020
A. Esmaeily, K. Kralevska, "Small-Scale 5G Testbeds for Network Slicing Deployment: A Systematic Review," Wireless Communications and Mobile Computing, 2021
A. Esmaeily et al., "A Cloud-based SDN/NFV Testbed for End-to-End Network Slicing in 4G/5G," 6th IEEE Conference on Network Softwarization (NetSoft), 2020.

Network slicing concept is important in 5G, even one factory may need slices with different requirements. Could you please add a paragraph reflecting your thoughts and finding in relation to potential use of WiFi-6 slicing?

The unit in Fig. 5 is not correct.

Author Response

Rev3 /1: Please improve the quality of the figures (too small and quite illegible). The figures were enlarged.

Rev3 /2: For Figures 3, 4, 5. The charts can be further improved in terms of the presentation of the results. Complete the names on the axes. The Figures 3-5 seem to be aligned with clear labels as they currently are.  

Rev3 /3: You can expand the summary of the paper and refer to other frequencies used in 5G. This topic has been extended in the Discussion and future work section.  

Rev3/4: There are several linguistic and stylistic errors. Please edit the text of the paper. We went through the paper carefully and corrected the grammatical errors found.

Round 2

Reviewer 2 Report

General comments:

All the recommended corrections indicated in the previous version of the review were taken into account. Thank you for introducing corrections and explanations to the paper in accordance with the reviewer's recommendations. I have no further comments.

Summary:

The presented description of an experiment will be a good contribution to the knowledge in area of wireless communications.

Reviewer 3 Report

The reviewer is satisfied with the current manuscript.